# Association between Geriatric Nutritional Risk Index and Post-Stroke Cognitive Outcomes

**DOI:** 10.3390/nu13061776

**Published:** 2021-05-23

**Authors:** Minwoo Lee, Jae-Sung Lim, Yerim Kim, Ju Hun Lee, Chul-Ho Kim, Sang-Hwa Lee, Min Uk Jang, Mi Sun Oh, Byung-Chul Lee, Kyung-Ho Yu

**Affiliations:** 1Department of Neurology, Hallym University Sacred Heart Hospital, Hallym University College of Medicine, Anyang 14068, Korea; minwoo.lee.md@gmail.com (M.L.); iyyar@hallym.or.kr (M.S.O.); ssbrain@hallym.or.kr (B.-C.L.); 2Department of Neurology, Asan Medical Center, University of Ulsan College of Medicine, Seoul 05505, Korea; 3Department of Neurology, Kangdong Sacred Heart Hospital, Hallym University College of Medicine, Seoul 05355, Korea; brainyrk@gmail.com (Y.K.); leejuhun@kdh.or.kr (J.H.L.); 4Department of Neurology, Chuncheon Sacred Heart Hospital, Hallym University College of Medicine, Chunchon 24253, Korea; gumdol52@naver.com (C.-H.K.); bleulsh@naver.com (S.-H.L.); 5Department of Neurology, Dongtan Sacred Heart Hospital, Hallym University College of Medicine, Hwaseong 18450, Korea; mujang@gmail.com

**Keywords:** geriatric nutritional risk index, nutrition, stroke, cerebral infarction, cognitive impairment

## Abstract

Background: It is not yet clear whether nutritional status is associated with post-stroke cognitive impairment (PSCI). We examined the geriatric nutritional risk index (GNRI) on the domain-specific cognitive outcomes 3 months after a stroke. Methods: A total of 344 patients with acute ischemic stroke were included for the analysis. The GNRI was calculated as 1.489 × serum albumin (g/L) + 41.7 × admission weight (kg)/ideal body weight (kg) and was dichotomized according to the prespecified cut-off points for no risk and any risks. The primary outcome was PSCI, defined as having adjusted z-scores of less than −2 standard deviations in at least one cognitive domain: executive/activation, memory, visuospatial and language. Multiple logistic regression and linear regression analyses were performed to investigate the association between the GNRI and cognitive outcomes. Results: Seventy (20.3%) patients developed PSCI 3 months after a stroke. The mean GNRI was 106.1 ± 8.6, and 59 (17.2%) patients had low (<98) GNRI scores. A low GNRI was independently associated with the PSCI after adjusting for age, sex, education, initial stroke severity, stroke mechanism and left hemispheric lesion (odds ratio, 2.04; 95% confidence interval, 1.00–4.14). The GNRI scores were also significantly associated with the z-scores from the mini-mental status examination and the frontal domain (β = 0.04, *p*-value = 0.03; β = 0.03, *p*-value = 0.03, respectively). Conclusions: A low GNRI was independently associated with the development of PSCI at 3 months after an ischemic stroke. The GNRI scores were specifically associated with the z-scores of the global cognition and frontal domain cognitive outcomes.

## 1. Introduction

Malnutrition has been considered to adversely affect the prognoses of cardiovascular diseases [1,2,3,4] and vascular dementia [5]. Premorbid malnutrition is associated with multiple in-hospital complications, including infection, pressure sores and prolonged length of hospitalization after a stroke [6]. It is also known to predict short-term poor functional outcomes and mortality after an ischemic stroke [7]. While previous studies have failed to show the efficacy of early enteral tube feeding or nutritional supplementation added to a normal hospital diet in the context of both functional outcomes and mortality rate after a stroke [8,9], another study has shown that nutritional support is associated with improved global cognitive outcomes assessed by mini-mental state examination (MMSE) after a stroke [10].

Thus, as optimal nutritional intervention may improve cognitive performance in malnutritional stroke patients, it is crucial to assess the premorbid nutritional status following an ischemic stroke. Recently, several different screening tools have been used for the assessment of the nutritional status in stroke patients including the Mini Nutritional Assessment Short-Form (MNA-SF) and the Malnutrition Universal Screening Tool (MUST) [11,12,13]. However, it is not easy for stroke patients and caregivers to complete a structured questionnaire or describe accurately whether they experienced psychological stress or weight loss. The geriatric nutritional risk index (GNRI) has been proposed and used as an objective indicator of the nutritional status based on serum albumin levels and body indices in surgical patients [14], and in patients with cardiovascular disease [3] and malignancy [15]. Unlike other indices, the GNRI can be calculated without the patient’s cooperation or a nutritional specialist, thereby guaranteeing the generalizability of the research and clinical use [16]. The GNRI can be used immediately at stroke centers as it can be derived only from the information collected during acute stroke management without any additional resource utilization. The GNRI has already been validated for the prediction of outcomes after a cardiovascular disease, and a recent study has revealed its prognostic value as regards short-term functional outcomes after a stroke [7]. It has also been validated via correlation with traditional nutritional indices, including the MNA-SF and the MUST in stroke patients [13].

While the GNRI can be readily utilized in the real-world setting, the usefulness of the GNRI for the cognitive outcomes after a stroke has not yet been elucidated. Furthermore, previous studies only evaluated a brief screening test such as the MMSE without detailed cognitive tests in a limited number of patients [10]. Identifying domain-specific cognitive impairment due to malnutrition is also important for revealing the underlying pathophysiology and selecting appropriate treatment targets.

Thus, the primary aim of this study is to determine the association between premorbid nutritional status, assessed via the GNRI score, and the development of post-stroke cognitive impairment (PSCI) 3 months after an acute ischemic stroke. The secondary aim is to assess the association between the GNRI and both the global and domain-specific cognitive outcomes after a stroke.

## 2. Materials and Methods

### 2.1. Study Design and Population

This study is a retrospective analysis of a prospective stroke registry database [17] and post-stroke neuropsychological battery database in a tertiary university hospital. The study was approved by the Institutional Review Board of Hallym University Sacred Heart Hospital (IRB No.2021-03-008). Patient consent was waived due to the retrospective nature of the study and the minimal risk that it posed to the subjects.

From January 2011 to December 2015, participants who had suffered from ischemic stroke were consecutively enrolled from a prospective stroke registry in a tertial university hospital. We included patients with (1) a diagnosis of acute ischemic stroke who were admitted within 7 days of symptom onset and (2) available data from a neuropsychological test battery performed 3 months after ischemic stroke. We excluded participants with pre-stroke cognitive impairments who had previously been diagnosed with dementia or a premorbid modified Rankin score of >2. Finally, a total of 344 patients were included in the analysis.

### 2.2. Clinical Information

Demographic factors including age, sex and years of education were collected. Baseline characteristics, including a history or new diagnosis of hypertension, diabetes, hyperlipidemia, atrial fibrillation, smoking status and previous stroke or transient ischemic attack, were also collected. The evaluation of stroke characteristics included initial stroke severity assessed via the National Institute of Health stroke scale (NIHSS), and stroke subtypes according to the Trial of ORG 10,172 in Acute Stroke Treatment (TOAST) classification. The participants’ weight (kg) and height (m) were measured on the first day of admission with an automatic weight and height measuring machine (stadiometer BSM-330; InBody, Seoul, Korea). Those who could not stand alone were weighed in the supine position using an electronic bed scale (SCB330-7; SOHWA Inc, Seoul, South Korea). Laboratory information, including serum albumin level, was collected upon admission to the emergency department. We performed brain magnetic resonance imaging (MRI) on all participants with a 3Tesla whole-body MRI system (Achieva, Philips Healthcare, Best, The Netherlands). Diffusion-weighted images were obtained to define acute ischemic stroke. The lesion location and number were assessed and quantified as follows: (1) presence of left hemispheric lesions, (2) cortical vs. subcortical lesions only and (3) single vs. multiple lesions, which are known from previous studies to independently affect post-stroke cognitive outcomes.

### 2.3. Nutrition Status Assessment

Premorbid nutritional status was assessed using the GNRI, which was calculated as follows: 1.489 × serum albumin (g/L) + 41.7 × admission weight (kg)/ideal body weight (kg). The Lorentz formula was used to calculate ideal body weight according to the patients’ height and sex, as follows: [height (cm) − 100 − {(height (cm) − 150)/4}] for men and [height (cm) − 100 − {(height (cm) − 150)/2.5}] for women [18]. The study population was dichotomized into two groups, based on the original description of the GNRI, as follows: the low GNRI group (any nutritional risks; GNRI < 98) and the high GNRI group (no nutritional risk; GNRI ≥ 98) [16,19].

### 2.4. Neuropsychological Outcome Variables

We evaluated the 3-month cognitive outcomes using the Korean version of MMSE (K-MMSE) and the Korean version of the vascular cognitive impairment harmonization standards—neuropsychological protocol (K-VCIHS-NP). The K-VCIHS-NP comprises all four major cognitive domain evaluation tests: memory, executive/activation, language and visuospatial functions [20,21]. The neuropsychological tests were described in detail in our previous study [21]. Both the K-MMSE and the K-VCIHS-NP were translated into z-scores with adjustments for age, sex and years of education. The primary outcome was the development of PSCI, which was defined as the patient having a z-score for at least one cognitive domain below −2 standard deviations [22]. The secondary outcome was the z-score for each of the four cognitive domains of K-VCIHS-NP and K-MMSE.

### 2.5. Statistical Analysis

All the statistical analyses were conducted using SPSS version 26 (SPSS, Inc., Chicago, IL, USA) and R (version 4.0.3; R Foundation for Statistical Computing); two-tailed *p*-values of <0.05 were considered statistically significant. The baseline characteristics and the GNRI scores were compared between the no cognitive impairment group (NCI) and the PSCI group. The Shapiro–Wilk test for normality was performed for all continuous variables. The normally distributed data were compared using the Student’s *t*-test, and non-normally distributed data were compared using the Mann–Whitney U test for continuous variables. The non-normally distributed variables were presented as the median and interquartile range, and the normally distribute variables were presented with mean and standard deviation. For categorical variables, the chi-squared test or Fisher’s exact test were used as appropriate. We planned to exclude the variables with more than 5% missing values in the main analysis.

The differences in each cognitive outcome according to the GNRI group were analyzed by the Student’s *t*-test. For the primary outcome, the effects of low GNRI on the development of PSCI were investigated using logistic regression analyses. Covariates with a *p*-value < 0.10 in the univariate analysis and clinically significant variables, including age, sex, years of education and initial NIHSS, were adjusted for the multivariate analysis [23]. For the secondary outcome, multiple linear regression was performed to assess the contribution of GNRI scores to each cognitive outcome z-score, with adjustments for age, sex, education years and initial NIHSS.

## 3. Results

A total of 355 patients who met the eligibility criteria were enrolled in the study. Among them, seven patients were excluded due to pre-stroke cognitive impairment and a further four patients were excluded due to missing data for height and weight. Ultimately, a total of 344 patients was included in the final analyses. The mean age was 63.0 ± 12.0, and the median initial NIHSS score was 2 (interquartile range: 1–4). The mean GNRI was 106.1 ± 8.6, and 59 (17.2%) patients had low GNRI scores. Among the 344 patients, 70 patients (20.3%) developed PSCI 3 months after a stroke.

Table 1 shows the baseline characteristics between the NCI and PSCI groups. As shown in the univariate analysis, the PSCI group tended to be older and female, and had a lower frequency of suffering from hyperlipidemia. The PSCI group had a significantly higher frequency of patients with a low GNRI, and a higher initial NIHSS score. The stroke mechanisms according to the TOAST classification were significantly different between the two groups, with less frequent small vessel occlusion in the PSCI group. 

Comparing the z-scores of the K-MMSE and each cognitive domain revealed that the z-scores of the K-MMSE, executive/activation and language were significantly lower in the low GNRI group compared with those in the high GNRI group (Table 2). In the multiple logistic regression analysis, a higher risk of premorbid malnutrition (low GNRI) was significantly associated with an increased risk of the development of PSCI after adjusting for age, sex, years of education, initial stroke severity, TOAST classification and left hemispheric lesions (low GNRI crude odds ratio (OR), 2.66; 95% confidence interval (CI), 1.44–4.93; *p*-value = 0.002, adjusted OR, 2.04; 95% CI, 1.00–4.14; *p*-value = 0.049; Table 3) In the multiple linear regression analysis, the GNRI score was significantly associated with the K-MMSE and executive/activation domain z-scores after adjusting for age, sex, years of education and initial stroke severity (β = 0.04, *p*-value = 0.03 for K-MMSE, and β = 0.03, *p*-value = 0.03 for the executive/activation domain; Figure 1).

## 4. Discussion

In this study, we revealed two important findings. First, increased premorbid nutritional risks, represented as low GNRI scores, could act as a novel predictor for the development of PSCI 3 months after a stroke. Second, low GNRI scores were associated with poor cognitive performance in the K-MMSE and executive/activation domains. These results imply that premorbid malnutrition may be a potentially modifiable risk factor in an acute stroke period for the cognitive performance 3 months after a stroke.

### 4.1. GNRI as a Predictor for PSCI

In several studies, premorbid malnutrition was associated with initial stroke severity and functional outcome following an acute ischemic stroke [7,24,25]. According to the original study by Bouillanne [16] that established GNRI scores, it was suggested that the GNRI scores are categorized into four risk groups (very low-risk, GNRI ≥ 98; low-risk, 98 > GNRI ≥ 92; moderate risk, 92 > GNRI ≥ 82; severe risk, 82 > GNRI). A recent study in a stroke cohort revealed that the adjusted odds ratio for poor short-term functional outcomes in each GNRI group was 3.838 in the severe risk group and 1.522 in the moderate risk group [7]. As our study cohorts are composed of relatively mild stroke patients, those at severe or moderate risk were few in number. Thus, we combined all three risk groups and compared them with the very low-risk group, revealing that any premorbid nutritional risk is associated with poor cognitive performance after a stroke.

Our study suggests that the nutritional status evaluated in the acute stroke stage might affect the cognitive outcome 3 months after a stroke. The findings are consistent with previous studies that investigated the association between the MNA-SF and both the MMSE and Functional Independence Measure cognitive subscale. They revealed that malnutrition according to the MNA-SF was associated with poor cognitive improvement after an ischemic stroke [26]. Furthermore, another study with calorie-protein supplements evaluated the treatment effect in 3-week intervals from admission and revealed that the optimal nutritional supplement could prevent cognitive deterioration after a stroke [10]. The GNRI is of special interest for patients with potential cognitive impairment, as it only requires weight, height and serum albumin levels, while other indices such as the MNA-SF or MUST often require the completion of a structured questionnaire. Moreover, the combination of serum albumin levels with weight and height may reduce the potential confounding effects of variations in serum albumin levels due to hydration status or acute inflammation [16].

### 4.2. Pathophysiological Link between Malnutrition and Post-Stroke Cognitive Outcomes

There are several potential pathological mechanisms underlying the link between malnutrition and cognitive performance after a stroke, including its direct impact on the stroke severity, and the premorbid vulnerability of specific brain structures. Malnutrition adversely affects brain plasticity, and prevents proper protein synthesis and glucose utilization at the ischemic penumbra, which may lead to a more severe stroke and the worsening of symptoms [27]. Further, malnutrition often results in an immunosuppressive state, potentially leading to infection in the acute stroke period [28]. Malnutrition is also associated with an increased risk of white matter hyperintensities [29], microbleeds and mesial temporal lobe atrophy in the general population, and in patients with mild cognitive impairment or dementia [30]. Therefore, those with premorbid malnutrition may be more vulnerable to ischemic injury if they have an underlying amyloid-beta-related pathology or vascular burden.

In this regard, nutritional supplementation showed a beneficial effect on cognitive recovery after a stroke [10]. It was hypothesized that an increased supplement of amino acids could facilitate the activation of neural protein synthesis, thereby increasing the production of new cortical connections and axonal sprouting. These changes may both be associated with improved neural plasticity in the subacute rehabilitation stage of stroke. Increased synthesis of a neurotransmitter could be another protective mechanism by which nutritional support with proteins may enhance the cognitive recovery after a stroke [10].

Various biomarkers including brain-derived neurotrophic factor (BDNF), insulin-like growth factor 1 (IGF-1), interleukin 6 and 10 and microRNA-132 (miR-132) were suggested to surrogate these associations [31]. Among them, the BDNF is abundant in brain areas and involved in neural plasticity and neurogenesis. As malnutrition is associated with a lower level of circulating BDNF, premorbid malnutrition may hinder the process of cognitive recovery after a stroke [31]. The IGF-1 in an acute stroke was associated with neurological recovery [32], while the miR-132 was negatively associated with Montreal cognitive assessment scores [33]. Furthermore, a low body mass index has been associated with the cortical thinning of the middle frontal cortex and the frontal pole in a general elderly population [34], which may partially explain the domain-specific occurrence of executive/activation dysfunction observed in the low GNRI group in our study. However, in our study, these biomarkers were not analyzed together. Further analysis using serum and imaging biomarkers is required to determine the pathophysiological association between a low GNRI and domain-specific cognitive dysfunction more specifically.

### 4.3. Limitations

There are several limitations to our study. First, we only evaluated the GNRI according to the data derived on admission and did not assess further changes in the GNRI. Therefore, we were not able to consider the variation in the nutritional status that occurred between admission and the first 3-month period following a stroke. Secondly, we did not assess other surrogate indexes of the nutritional status, such as the MUST or MNA-SF, and thus were unable to compare these with the GNRI. Both the MUST and MNA-SF, along with other nutritional markers, mostly require questionnaires from patients, and thus could only be prospectively collected. Third, patients suffering from aphasia or severe stroke who were not able to undergo neuropsychological tests were excluded; hence, most of the participants in our cohort had a mild stroke severity and/or mild nutritional risks, which precludes the generalization of our findings to all stroke patients.

## 5. Conclusions

In conclusion, the risks from premorbid nutrition, represented as low GNRI scores, were independently associated with PSCI at 3 months after the occurrence of ischemic stroke, with the specific occurrence of global and frontal dysfunction. Considering that the premorbid nutritional status is an important modifiable risk factor that may determine cognitive performance after a stroke, the GNRI scores may be utilized as a readily available screening tool for malnutrition and used as a predictor of PSCI after a stroke.

Future studies investigating the impact of nutritional intervention on the domain-specific cognitive outcomes after an ischemic stroke, who are at risk of malnutrition, are needed to identify the causal relationship between them and to provide evidence for guidelines of acute stroke management.

## Figures and Tables

**Figure 1 nutrients-13-01776-f001:**
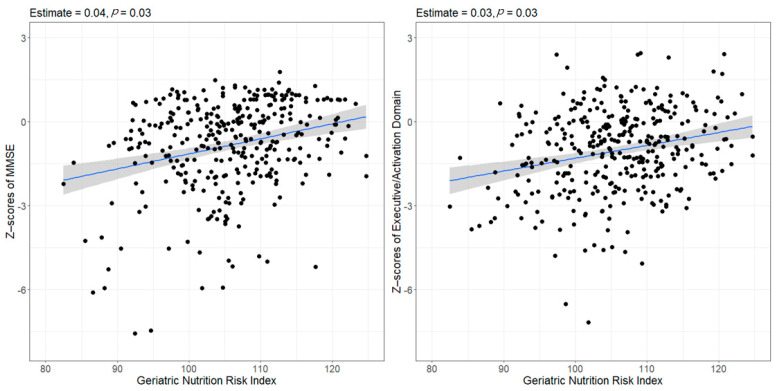
Scatter plots indicating the relationship between the GNRI and cognitive domain z-scores. NRI was significantly associated with the z-scores of the MMSE and the executive/activation domain via multiple linear regression with adjustments for age, sex, education and initial NIHSS scores. Abbreviations: GNRI—geriatric nutritional risk index; MMSE—mini-mental status examination; NIHSS—National Institute of Health stroke scale.

**Table 1 nutrients-13-01776-t001:** Baseline characteristics according to the presence of post-stroke cognitive impairment.

	NCI (*n* = 274)	PSCI (*n* = 70)	*p*-Value
Low GNRI (*n*, %)	38 (13.9%)	21 (30%)	0.003
GNRI	106.9 ± 8.1	102.9 ± 9.6	0.001
Age	62.0 ± 12.0	66.9 ± 11.3	0.003
Sex, male	184 (67.2%)	37 (52.9%)	0.037
Years of Education	9.6 ± 5.4	8.7 ± 4.9	0.233
Vascular Risk Factors			
Hypertension	151 (55.1%)	45 (64.3%)	0.212
Diabetes mellitus	63 (23.0%)	22 (31.4%)	0.192
Hyperlipidemia	63 (23.0%)	8 (11.4%)	0.049
Smoking history	124 (45.3%)	32 (45.7%)	0.999
Atrial fibrillation	6 (2.2%)	4 (5.7%)	0.243
Coronary artery disease	14 (5.1%)	5 (7.1%)	0.710
IQCODE	3.2 ± 0.4	3.3 ± 0.4	0.077
Initial NIHSS	2.0 (1.0; 3.0)	3.0 (1.5; 7.5)	0.001
Stroke Subtype (TOAST)			0.002
LAA	92 (33.8%)	35 (50.7%)	
SVO	142 (52.2%)	20 (29.0%)	
CE	12 (4.4%)	8 (11.6%)	
OD + UD	26 (9.6%)	6 (8.7%)	
Lesion Locations			
Subcortical lesion	51 (18.6%)	11 (15.7%)	0.697
Left hemispheric lesion	75 (27.4%)	28 (40.0%)	0.056
Multiple lesions	13 (4.7%)	5 (7.1%)	0.615

Abbreviations: NCI—no cognitive impairment; PSCI—post-stroke cognitive impairment; GNRI—geriatric nutritional risk index; IQCODE—informed questionnaire on cognitive decline in the elderly; NIHSS—National Institute of Health stroke scale; TOAST—Trial of ORG 10,172 in Acute Stroke Treatment; LAA—large artery atherosclerosis; SVO—small vessel occlusion; CE—cardioembolism; OD—other determined; UD—undetermined.

**Table 2 nutrients-13-01776-t002:** Comparison of z-scores of K-MMSE and each cognitive domain.

Cognitive Domains	High GNRI Group (*n* = 285)	Low GNRI Group (*n* = 70)	*p*-Value
K-MMSE	−0.9 ± 1.9	−1.4 ± 2.1	0.043
Memory	−0.9 ± 1.3	−1.2 ± 1.3	0.080
Executive/Activation	−0.1 ± 1.5	−1.6 ± 1.5	0.001
Visuospatial	−1.2 ± 1.9	−0.5 ± 1.3	0.573
Language	−0.9 ± 1.3	−0.5 ± 1.3	0.006

Student’s *t*-test was performed. Abbreviations: K-MMSE—Korean version of mini-mental status examination.

**Table 3 nutrients-13-01776-t003:** Multivariable analysis for possible predictors of PSCI.

	Crude OR (95% CI)	*p*-Value	Adjusted OR (95% CI)	*p*-Value
Low GNRI	2.66 (1.44–4.93)	0.002	2.04 (1.00–4.14)	0.049
Age (per 10 year increase)	1.42 (1.13–1.78)	0.003	1.38 (1.04–1.83)	0.027
Male sex	0.55 (0.32–0.93)	0.027	0.60 (0.30–1.19)	0.145
Education	0.97 (0.92–1.02)	0.232	1.05 (0.98–1.12)	0.186
Initial NIHSS	1.13 (1.06–1.19)	<0.001	1.10 (1.03–1.18)	0.006
TOAST	Ref. SVO			
LAA	2.70 (1.47–4.97)	<0.001	2.07 (1.06–4.01)	0.032
CE	4.73 (1.72–12.99)	0.001	2.03 (0.64–6.44)	0.978
OD	1.18 (0.14–10.34)	0.879	1.47 (0.16–13.72)	0.766
UD	1.77 (0.60–5.26)	0.300	1.64 (0.52–5.17)	0.700
Left hemispheric	1.77 (1.02–3.06)	0.041	1.68 (0.92–3.06)	0.089

Adjusted for GNRI, age, sex, hyperlipidemia, years of education, initial NIHSS, stroke subtype (TOAST) and left hemispheric lesion. Abbreviations: OR—odds ratio; CI—confidence interval. Otherwise, refer to Table 1.

## Data Availability

The data are available from the corresponding author on reasonable request.

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
