# Peer review of "Association between Geriatric Nutritional Risk Index and Post-Stroke Cognitive Outcomes"

_nutrients, 2021, doi:10.3390/nu13061776_

Round 1
Reviewer 1 Report
The work is excellently presented with minor corrections, as shown below:
Line 92: The identifying numerical for the phrase "presence of left ventricular lesions" should be (i) instead of (ii).
Line 110-111: Please site the 'previous study' mentioned in the sentence.
Line 107 and 111: The abbreviation K-MMSE is used in line 111, but merely MMSE is defined in line 107.
Lines 128 and 149: Although the abbreviations, NIHSS and TOAST, are defined in the abstract, it would better to redefine them when they are first used in the body of the article, on pages 128 and 149).
Reviewer 2 Report
- Abstract: Clarification is needed. Abstract needs to focus on what was done, what was found and what was concluded. The statistical tests conducted is unclear in abstract. The outcome measures are unclear as outcome measures were only introduced in the results.
- Line 44 and Line 45: What do you mean by 'proper' supplements/supplementation. Please consider rewording.
- Introduction does not provide a review of literature. Major improvement is needed to search the literature and introduce, nd critically appraise the body of knowledge.
- Lines 50-53: Why amongst many screening tools GNRI was selected? Example of other screening tools need to be introduced and the reason for selecting GNRI need clarification.
- The Rationale needs major clarification. It is not clear why this study is needed, why this would be a logical continuation of the previous studies and how this addresses the shortcomings of the previous studies.
- Line 59, why you have hypothesised so? The observational studies do not test theory, so how can you test the hypothesis?
- Line 61, the aim need clarification. if there are more than one aim, you can refer to them as primary and secondary aim.
- Lines 66-69, clarification is needed if this is an observational study or a retrospective reanalysis of the patients' data?
- What was the timeframe of the study?
- Patient consent was waived, this is in opposition to Helsinki principles. Please attach a copy of your ethical approval, with official translation for the reassurance of the journal.
- Line 70-77, age is a major determinants of the variables of this study. What was the inclusion criteria for age, and why?
- Lines 66-77, what is sample size calculation and why?
- Line 78, the heading is unconventional. Please consider rewording.
- Line 90, MRI machine characteristics need clarification.
- Lines 97-98, please provide reference.
- Line 99-100, please provide reference.
- Study design: please clearly comment what are you dependent and independent variables. Also, what are your primary, secondary and tertiary outcome measures?
- Lines 82-84, please elaborate what are these and provide reference for each.
- Please provide information what was the participants pool, how many included, excluded and why?
- Lines 119-124, please provide clarification how normal distribution was tested, what variables were non-normally distributed, what actions were taken to provide outliers or transform the data, before reporting choice of parametric and non-parameteric statistics.
- Please provide ROC and Area Under Curve analysis, to confirm/strengthen the findings of the logistic regression.
- Line 144-146, if this is part of the results, please clarify how the mechanisms were different and what is the evidence for the statement.
- Table 1, why some values in red?
- Table 1, please improve the table presentation (e.g. what is criteria for education, hypertension, arterial fibrillation, and many abbreviations)
- Results, please provide effect size throughout.
- Discussions, please first of all (first paragraph) comment on the key findings, how do they link with the aim, and what findings mean.
- Figure 1, please move to the results section.
- Lines 217-19, please elaborate on limitations. For instance, why MUST, MNA-SF have not been used?
- Please provide a clear statement for the direction of the future research.
Reviewer 3 Report
The paper is a very interesting article testing the association between nutritional status and post-stroke cognitive outcomes.
The authors evaluated a large population of ischemic stroke patients estimating the Geriatric Nutritional Risk index and cognitive function.
They founded that post-stroke cognitive impairment is associated with a low level of Nutritional geriatric index via specific involvement of the frontal domain of cognition.
The topic is exciting because it shines the light on the metabolic changes that develop in the brain affected by stroke.
It is also essential to recognize that premorbid malnutrition can damage the brain's repair capabilities.
In my opinion, the introduction is clear enough.
The material and methods section is written correctly.
The results could better described cognitive performance showing a table with the MMSE scores.
The discussion could be expanded by introducing aspects of pathophysiology concerning the protective factors of nutrition.
Furthermore Authors may consider other biomarkers from Post Stoke Cognitive Impairment and evaluate if there is a connection with nutritional risk. (e.g: Post-Stroke Cognitive Impairment: A Review Focusing
on Molecular Biomarkers)
Reviewer 4 Report
This paper is well constructed, the methods and results are clearly written.
And as it stands it is well laid out.
However, there are several minor issues that can be addressed to improve the paper.
Line 63: PSCI : Acronym?? First time the word has been used in the introduction section
Line 111: in our previous study (citation?)
Line 123: Mann-Whitney U test for continuous variables?
Line 149: Years of education---if there is no significant association from univariate analysis, then is it necessary to add in the model?
Reduce use of conjunctive verb such as “as such”
Introduction: Add a few sentences for choosing 3 months? (Is it chosen randomly?)
Methods: IRB number?
Effect size?
Is the sample powered to observe the difference?
Table 1 and Table 2: PSCI (n = 70) Vs NCI (n = 274): Justification to observe to true difference for the sample size differences
Table 1: Can be merged CE, OD, and UD into one group
Discussion: Need a more comparison with results from other studies
Thank you for the opportunity. Best wishes.
Round 2
Reviewer 2 Report
Thank you for making extensive amendments and providing an extensive and clear response and clarification.